# What Is Hiding in the Israeli Mediterranean Seawater and Beach Sand

**DOI:** 10.3390/jof8090950

**Published:** 2022-09-10

**Authors:** Michael Frenkel, Hanan Serhan, Shlomo E. Blum, Marcelo Fleker, Edward Sionov, Sharon Amit, Zeela Gazit, Shiraz Gefen-Halevi, Esther Segal

**Affiliations:** 1Department of Clinical Microbiology, Sackler School of Medicine, Tel Aviv University, Tel Aviv 562000, Israel; 2Department of Clinical Bacteriology and Mycology, The Kimron Veterinary Institute, Bet Dagan 50200, Israel; 3Institute for Post Harwest and Food Sciences, Agricultural Research Organization, The Volcani Center, Rishon LeZion 7528809, Israel; 4Microbiology Laboratory, Sheba Medical Center, Tel Hashomer 5262000, Israel

**Keywords:** fungi, beach sand, seawater

## Abstract

Objective: In the present study, we aimed to investigate the presence of fungi that may affect human health in sand and water on Israeli Mediterranean Sea coast beaches. Methods: The study included screening of the sand and water of six urban beaches from north to south on the Israeli Mediterranean coast. Sand samples were extracted with water, and the water wash was cultured and quantitated. Water samples were quantitated as well. MALDI-TOF MS analysis and ITS sequencing identified the fungi. Results: The study considered several parameters: 1. Presence of fecal-contamination-related fungi; 2. Presence of dermal-infection-related fungi. 3. Presence of allergy-related fungi; 4. Presence of fungi posing risk for immunocompromised individuals. The screen revealed that about 80% of the isolates were molds and about 20% yeasts. The mold species included opportunistic pathogens and potential allergens: *Aspergillus fumigatus* and other *Aspergillus* species, *Fusarium*, *Penicillium,* and *Mucorales* species. Yeast isolates included *Candida*—including the human commensals *Candida albicans* and *Candida tropicalis*—*Cryptococcus,* and *Rhodotorula* species. Conclusions: The results suggest that beaches should be monitored for fungi for safer use, better management, and the benefit of public health.

## 1. Introduction

Studies on the microbial flora of beach sand and water of various recreational water bodies around the globe have shown that a variety of microbial species can be detected, including bacteria and fungi [1,2,3]. Many of these environmental microbes have a capacity to cause human infections [4,5,6], some of which may be of great significance for human health, such as the recent discovery of the antifungal resistant *Candida auris*, isolated from a marine ecosystem [7]. Earlier studies have shown the importance of both bacterial [5,6] and fungal flora from marine sources or freshwater bodies, such as lakes or rivers, in the context of human health.

Moreover, climatic changes due to global warming may result in changes in microbial communities [8,9] and result in greater global distribution of microbial species from one geographic area to another. This phenomenon is demonstrated by the distribution of *C. auris* from its original habitat to other habitats [8].

The finding of harmful microbial flora in such environmental niches leads to the development of regulatory measures for recreational water and sands regarding contamination by bacterial species. The regulations include criteria to assess the presence of fecal contamination of sand and water via quantitation of *Escherichia coli* and Enterococci in these environmental niches [10] or other methodology [11].

Research regarding fungal flora in the sands and water of recreational water bodies regarding effects on human health is less advanced than that regarding bacterial flora. Nevertheless, such studies have been reported regarding fungal flora found in various geographic areas, such as those of Sabino et al. [5] in beaches around mainland Portugal and its insular regions. These studies have shown the presence of Dermatophytes—fungi known as agents of cutaneous infections—in sand in addition to other fungal species [6].

A previous survey assessed the fungal flora in the sand of six beaches of the Israeli Mediterranean coast during the four seasons of a year [12]. Such preliminary exploratory survey corroborated studies from other geographic areas [13,14], revealing the presence of fungi in sand. However, that study did not explore the flora in the seawater nor the quantitative aspects of the contaminating fungal flora comprised of various yeast or mold species.

In line with the previous study, the present work focuses both on sand and water, as well as on quantitative aspects, by estimating the total load of the fungal flora in sand and water as well as the quantities of the specific fungal species. In addition, detailed qualitative and quantitative data regarding the various fungi isolated at the different areas are described. Such data could be a basis for regulatory measures and may have implications for public health. The present study can improve the current knowledge about the human health impact of such a particular marine ecosystem.

In the present study, fungal sand and water contamination was assessed in respect to possible impact on human health and well-being, particularly in reference to three aspects: (1) fecal contamination, as judged by the presence of the human gastrointestinal (GI) commensal, *Candida albicans*, other *Candida* spp., and *Rhodotorula* spp.; (2) contamination by fungi known for the involvement in dermal infections, such as dermatophytes; (3) presence in sand of various molds, such as *Aspergillus*, *Penicillium*, *Fusarium*, *Mucorales,* or other molds which may be the cause of respiratory allergies as well as a potential risk factor for invasive infections in immunocompromised individuals.

The following manuscript reports the findings in sand and water explored during a two-year survey of six beaches of the Israeli Mediterranean coast.

## 2. Materials and Methods

Fungal sand and water contamination was assessed in 6 urban beaches on the Israeli Mediterranean coast, as in the previous study [12], from north to south: Haifa, Keisaria, Tel Aviv, Palmachim, Ashdod, and Ashkelon.

These beaches were chosen as they cover most of the Israeli Mediterranean coast. Four are beaches in large cities (Haifa, Tel Aviv, Ashdod, and Ashkelon), and the other two (Keisaria and Palmachim) serve as bathing sites for a large inland population. Two of the beaches (Haifa and Ashdod) are major ports in the north (Haifa) and south (Ashdod) of the Israeli coast.

Sampling was performed during the bathing season and pre/post-bathing season. Sand and water were collected 3 times at each beach.

### 2.1. Sand Sampling

Based on the studies of Sabino et al. [15], dry sand was collected about 10 cm deep from the middle of the dry sand of shore in areas where bathers lie down. The distance of the collection area from the shore is dependent on the topography of the specific beach. The sand was collected with sterile gloves into sterile plastic bags. All beaches screened are urban beaches with high human use. All repeated collections (tree) were performed at the same locations. Sand samples were brought to the laboratory and kept at 4 °C until processed—up to 5 days. A total of 40 g of the collected sand was extracted under shaking (100 rpm) using 40 mL of sterile distilled water for 30 min.

The sand wash was diluted (1:10; 1:100), and the diluted and undiluted wash was plated (0.2 mL) on Sabouraud’s dextrose agar (SDA) plates supplemented with chloramphenicol. The agar plates were incubated at 28 °C and 37 °C and monitored for growth for up to 3 weeks. The fungal load in sand was evaluated quantitatively by enumeration of colony-forming units (CFU)/gram of sand.

### 2.2. Water Sampling

An amount of 0.2 mL of water was plated on SDA plates and incubated at 28 °C and 37 °C and monitored for growth for 3 weeks. The fungal load in water was evaluated quantitatively via enumeration of colony-forming units (CFU)/mL of water.

### 2.3. Fungal Identification

The fungi were identified phenotypically and spectrally using matrix-assisted laser/desorption ionization mass spectrometry (MALDI-TOF MS) using the Bruker system (with two databases) and by sequencing of the internal transcribed spacer region (ITS sequencing) (not all isolates).

### 2.4. Phenotypic Identification

Phenotypic identification was performed by standard mycological assays [16], in many instances only until the stage of differentiation between yeasts and molds.

### 2.5. MALDI-TOF MS

All isolated fungi were identified using MALDI-TOF MS. Isolated fungal colonies on SDA plates were identified using MALDI-TOF MS. Samples were prepared following the protocol for fungi identification as by the manufacturer. Spectra were obtained with an Autoflex system (Bruker, Leipzig, Germany). Identification was performed using the MBT software, with the latest upgrade of the Bruker fungi library (MBT 7854 Species/Entry List September 2018). In parallel, spectra were also analyzed also using the MSI library [17] accessed online; https://biological-mass-spectrometry-identification.com/msi/-LIBRARY (7 September 2022). Part of the fungi isolates were also analyzed using the MALDI-TOF-MS system in another medical center (Sheba Medical Center).

### 2.6. DNA Extraction, Amplification and Sequencing

As in the previous study [13], fungal genomic DNA was isolated from lyophilized mycelium/yeasts cells grown overnight in YPD medium. The DNA was extracted at 65 °C for 30 min, using a lysis buffer containing hexadecyltrimethyl-ammonium bromide (CTAB). The extracts were cooled prior to the addition of an equal volume of chloroform, gently mixed and centrifuged at 6000 rpm for 10 min. The aqueous supernatant was recovered, and the nucleic acids were precipitated with an equal volume of 2-propanol. The DNA was re-suspended in TE buffer solution (Tris EDTA, pH 8.0) containing RNase A at 10 µg/mL and further purified via phenol-chloroform extraction (A260/A280 ratio of 1.8–2.0). Finally, the DNA was precipitated with 100% ethanol containing 3 M sodium acetate, rinsed in 70% (*v*/*v*) ethanol, and resuspended in TE buffer. DNA quality and yield were determined using a NanoDrop One spectrophotometer (Thermo Scientific, Waltham, MA, US). The internal transcribed spacer regions, ITS 1-5-8 S and ITS 2 ribosomal DNA, were used to compare the ITS1-ITS2 nucleotide sequences. The universal ITS primers, ITS1 (5′-TCCGTAGGTGAACCTGCGG-3′) and ITS4 (5′-TCCTCCGCTTATTGATATGC-3′), were used, and the ITS regions were amplified selectively via PCR. The sequence of nucleotide alignments obtained was referenced against the NCBI database with the nucleotide BLAST program (https://blast.ncbi.nlm.nih.gov/Blast.cgi, accessed on 7 September 2022) [18].

## 3. Results

### 3.1. The Fungal Flora in Sand and Water

The screen of the six beaches of the Israeli Mediterranean Sea coast yielded a total of 232 fungal isolates.

As shown in Figure 1, the great majority of the fungal isolates, 85% (196 out of 232), were isolated from sand, while 15% were found in water.

Regarding the distribution of the fungal isolates in respect to the proportions of yeasts versus filamentous fungi, it can be noted that both sand and water showed that the majority of the isolated fungi were molds.

In sand, molds constituted 83% of the isolates (160/193), and in water, 92% (33/36) were molds.

This constant observation of greater contamination of both sand and sea water by molds than by yeasts should be taken into consideration when relating to possible relevance regarding human health and wellbeing.

### 3.2. The Yeasts in Sand and Water

As indicated afore, the majority of fungi isolated were filamentous fungi, namely molds, and the yeasts were less frequent in both niches—sand and water. The sand harbored the majority of the yeasts, namely 36 out of 39 isolates, corresponding to 92% of the total isolates. The speciation of the isolated yeasts revealed a minority in terms of number of isolates in comparison to molds. However, the isolates revealed significant diversity in terms of speciation (Figure 2). Most of the isolated yeasts, 31/39 (79%), have been assigned to the genus *Candida*. The other eight fungal isolates were taxonomically assigned to the genera *Cryptococcus* (3 isolates), *Rhodotorula* (3 isolates), *Kodamea* (one isolate), and *Collariella* (one isolate). The species diversity was more abundant in sand than in water.

### 3.3. The Molds in Sand and Water

As previously mentioned, the great majority of fungal isolates in sand and water were molds (193/232) (Figure 1). This observation is strengthened when separating the molds from the yeasts—193 vs. 39, respectively. The most common molds isolated were the Aspergilli (164/193) both in sand and water. It is of interest to note that from the water, only *Aspergillus* species and a single *Mucorales* species, *Rhizopus arhizus*, were isolated. Other mold isolates from sand included *Penicillia* and *Fusarium species* and single isolates of rarer molds (Figure 3). As presented in Table 1, the most common *Aspergillus* species isolated was *A. tubengensis*, with 47 isolates in sand and 13 in water. This was followed by another mold species of the *Aspergillus nigra* group: *A. welwetchiae,* with 31 and 10 isolates in sand and water, respectively.

### 3.4. Fungal Load in Sand and Water

To assess the fungal load in sand and water quantitatively, we carried out enumeration of colony forming units (CFU). The data are shown in the following panel (Figure 4) composed of six figures showing the CFU number of fungi in sand and water from each of the six beaches surveyed.

The CFU determination of fungi in the two systems—sand and water—strengthens the qualitative observation that sand is the medium that harbors higher levels of fungal units (CFU), i.e., higher fungal load. Thus, sand is of increased relevance as a source of potential human contamination by fungi than recreational seawater. Furthermore, although there are differences between the beaches, the fungal load is higher for molds in all screened beaches than for yeasts, which may indicate the increased relevance of human contamination by mold species over that of yeast species.

In this connection, it is of interest to note that in contrast to the general observation that sand harbors more fungi than water, the enumeration of fungal colonies revealed a slightly different observation in some instances. In two of the beaches (Haifa and Cesarea, Figure 4a,b), there were more CFU/mL of *Aspergillus welwitchiae* in water than in a gram of sand. Screens of two other beaches more to the south of the Israeli Mediterranean Sea coast, namely Palmachim and Ashdod (Figure 4d,f), showed a higher number of CFU/mL of *Aspergillus tubengensis* in water than in a gram of sand. Although both *Aspergillus* species belong to the same group, the Nigra group apparently has different behavior. This observation raises the possibility of differences in the beach sites in regard to composition of the beach sand. Composition of beach sand is one of the characteristics affecting the flora harbored in sand [19].

## 4. Discussion

As mentioned previously, the current study is a continuation of a previous smaller study [13] on the fungal flora of the Israeli Mediterranean Sea coast. Both sand and water were explored; the latter was not assessed in the previous study. We also quantitatively assessed fungal load, a measurement not performed in the former survey.

This study corroborated several findings of the former study, thereby leading to a greater validity of the observations. Furthermore, the findings of the Eastern Mediterranean Sea coast are an additional asset regarding fungal flora occurrence in various water bodies [19]. This information helped to define the number of fungal units in sand used as an indicator parameter for initiating regulatory measurements of beach sands.

The current findings revealed the presence of yeast species such as *Candida tropicalis* [20], a known human commensal which can be indicator of contamination by bathers (I), in sand and water. In contrast, the human commensal *Candida albicans* was isolated only from sand, possibly due to the dilution effect or higher intolerance of salinity of seawater. Two other Candida species found both in sand and in water, albeit at lower numbers in water, were *Candida guilliermondii* and *Candida metapsilosis*. These species are known as causative agents of invasive and mucocutaneous human infections [20]. *Candida guilliermondii* is known for its resistance to antifungal drugs [21].

In addition, other Candida species known as causes of human morbidity, particularly in immunocomprised individuals, such as *Candida parapsilosis, Candida lusitaniae*, and others (Figure 2), were also detected in sand. In a report by Segal et al. (2015) on onychomycosis in Israel [22], it was shown that *Candida parapsilosis* was the most frequent *Candida* species in affected nails.

Furthermore, sand also harbored *Cryptococcus* and *Rhodotorula* species as well as *Kodamea ohmeri,* all of which may cause human infections in susceptible populations [23,24,25,26,27]. *Cryptococcus albidus* is an environmental yeast known to cause infections in immunocompromised individuals [23,24]. A rarer *Cryptococcus* species, *Cryptococcus uzbekistanensis*, was reported as causing infection in an immunocompromised patient with lymphoma [25]. This rare yeast species is currently under investigation (NGS sequencing). In turn, *Kodamea ohmeri* is a new emerging agent of human morbidity [26,27]. In a recent publication, Zhou et al. [26] reported on 67 cases of systemic infections caused by *Kodamea ohmeri* in immunocompromised patients suffering from various comorbidities. Another systematic review on this fungus in 44 patients was reported by Ioannou et al. [27].

The molds found in the current study (Figure 1 and Figure 3 & Table 1), which constitute 85% of all fungi isolated, are composed primarily of *Aspergillus*, *Penicillium*, *Fusarium*, and *Mucorales* species and a number of single representatives of various other species. All of the above were isolated from sand. Water harbored only *Aspergillus* species and a single *Mucorales* species. Here too, as in the case of yeasts, sand seems to be a preferred medium for existence. Of note, the most frequently isolated Aspergillus species both from sand and from water was *Aspergillus tubingenensis,* followed by *Asppergillus welwitschiae*, *Aspergillus niger,* and *Aspergillus flavus (*Table 1). It seems that the Aspergilli of the Nigra group are more adaptable to different media than the other molds. Whether melanin might have a role can only be hypothesized [28]. An additional observation regarding the Nigra group is the different behavior in respect to quantitative assessment of fungal load in the two media—sand vs. water—as indicated in the “Results” section (Figure 4). Whether this is related to conidial size or conidial weight remains an open question. Both *Aspergillus* species may act as allergens, inducing respiratory hypersensitivity. Hence, presence of fungal spores at the beaches may be considered as a risk factor to allergy-prone individuals and possibly take part in ABPA (allergic bronchopulmonary aspergillosis) in prone individuals.

Another characteristic of diversity among the different Aspergilli was growth at 37 °C. While some of the *Aspergillus niger* isolates could grow at 37 °C, *Aspergillus weswitschiae* isolates could not. This finding may have significance, as the ability to grow at 37 ° C is one of the characteristics of potential pathogenicity in humans. Nevertheless, *Aspergillus welwitschiae* has been described in human infection [29].

It is of note that the major pathogenic *Aspergillus* species, *Aspergillus fumigatus* [30], was found both in sand and in water, albeit in the latter less frequently. This species, similarly to other *Aspergillus* species, may affect human health by acting as an allergen as well as via infection [31].

A major concern regarding *Aspergillus fumigatus* is the emergence of azole-resistant strains due to agricultural azole use [32,33].

The quantitation of the fungal load by enumeration of CFU showed some diversity among the different beaches, which may possibly result from various factors, such as the specific demography of the bathers. However, all screened beaches are urban beaches, and the proximity of two beaches (Haifa in North and Ashdod in the South) to port activity may have some impact on fungal quantity and species diversity. It is known that port sites may have spills of hydrocarbons, which would be suitable sites for Basidiomycetous yeast species, such as Rhodotorula, with ability to degrade such hydrocarbons [34].

Another parameter that was difficult to assess was the effect of season on fungal diversity and quantity in the two niches, sand and water. The reason being, as already noted in our first smaller study [13], that in Israel there, are no clear distinctions between the four yearly seasons: winter, spring, summer, and fall.

In summary:An extensive 2-year study that screened mycobiota in sand and water of the Israeli Mediterranean Sea coast revealed a great diversity of yeast and mold species, primarily in sand and to a minor extent in water.Many of the yeast and mold species have the potential to cause human disease, particularly in immunocompromised or debilitated individuals.In addition to the infectious process, fungi may cause human morbidity by provoking different types of allergies. Many of the mold species which are airborne may be associated with hypersensitivity reactions in allergy-prone bathers frequenting the recreational water bodies and the beaches around such water bodies.The data presented in this study may be a basis for regulatory measures to control the level of fungal mycobiota that occur in sand and seawater.Such regulatory measures are recommended for the benefit of public health.

## Figures and Tables

**Figure 1 jof-08-00950-f001:**
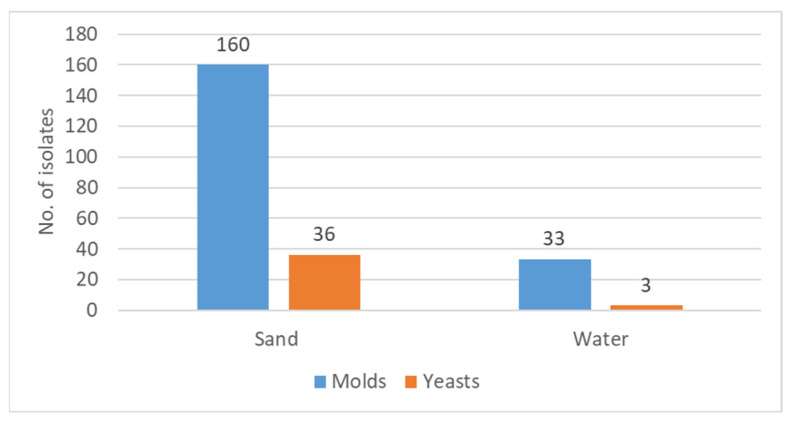
Total number of fungal isolates in sand and in water. The Y-axis represents the number of isolates of yeasts and molds in each medium.

**Figure 2 jof-08-00950-f002:**
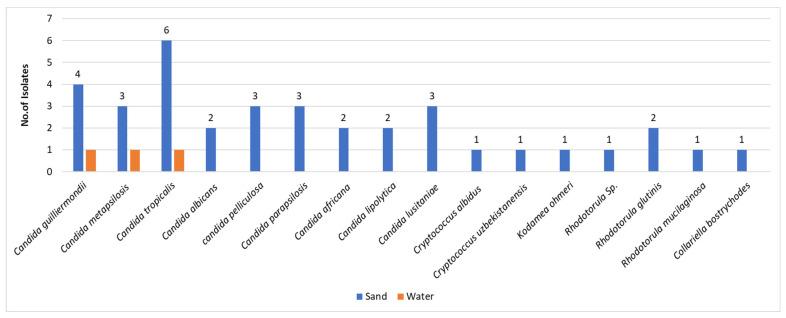
Number of isolates/yeast species in sand and water. The Y-axis represents the number of isolates.

**Figure 3 jof-08-00950-f003:**
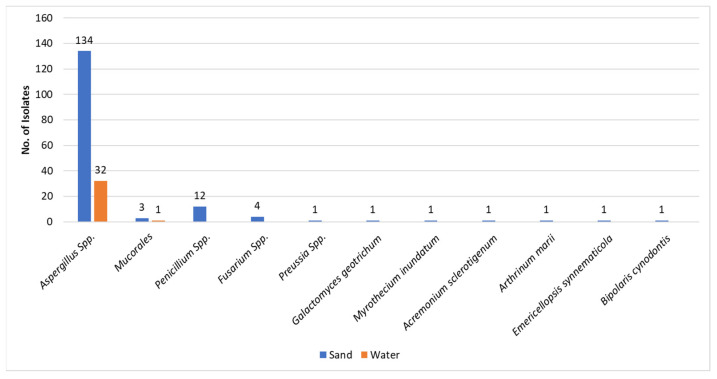
Molds in sand and in water. The Y-axis represents the number of isolates.

**Figure 4 jof-08-00950-f004:**
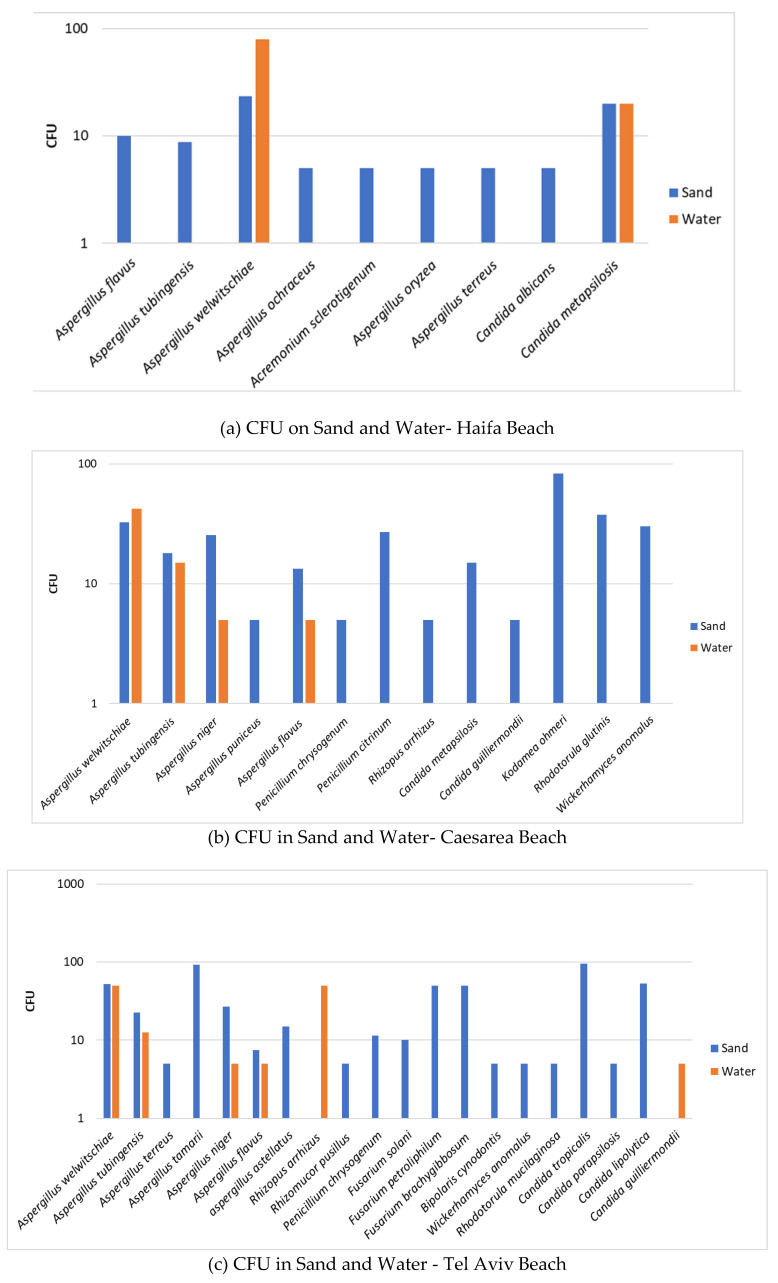
Fungal load in sand and water. Axis Y is CFU per gr for sand and CFU per ml for water. The error bars are S.D.

**Table 1 jof-08-00950-t001:** Number of isolates per mold species identified in water and sand.

Species	Molds in Water	Molds in Sand
** *Asp. tubingensis* **	13	47
*Asp. welwitschiae*	10	31
*Asp. niger*	4	17
*Asp. flavus*	4	17
*Asp. fumigatus*	1	5
*Asp. oryzea*		1
*Asp. astellatus*		1
*Asp. costaricensis*		1
*Asp. flavus\oryzae*		2
*Asp. luchuensis*		1
*Asp. puniceus*		1
*Asp. sydowii*		3
*Asp. tamarii*		1
*Asp. terreus*		4
*Asp. versicolor*		1
** *Rhizopus arrhizus* **	1	1
** *Mucor circinelloides* **		1
** *Rhizomucor pusillus* **		1
** *Penicillium chrysogenum* **		5
*Pen. citrinum*		1
*Penicillium/Paecilomyces*		1
*Pen. rubens*		2
*Pen. polonicum*		1
*Pen. thymicola*		1
*Pen. verrucosum*		1
** *Fusarium petroliphilum* **		1
*Fus. brachygibbosum*		1
*Fus. solani*		1
*Fus.* sp		1

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
