# Peer review of "What Is Hiding in the Israeli Mediterranean Seawater and Beach Sand"

_jof, 2022, doi:10.3390/jof8090950_

Round 1

Reviewer 1 Report

What Is Hiding In The Israeli Mediterranean Sea-Water And

Beach Sand

Frenkel, Michael; Serhan, Hanan; Blum, Shlomo; Fleker, Marcelo; Sionov, Edward; Amit, Sharon; Gazit, Zeela; Gefen-Halevi, Shiraz; Segal, Esther

Introduction

1. Weak and careless presentation: commensal, Candida albicans, other Candida; with taxon names not properly italicised.

2. Marine fungi growing on sand and in seawater are well documented, yet there is no mention of this in the introduction. Also, the website marinefngi.com lists faecal contamination by marine yeasts or those that cause human gastrointestinal infection.

Materials and methods

Details of how the water samples were made is not clear. Were samples collected in the intertidal? and how was the water obtained? Why were the wate samples not diluted as in the case of the sand? Was chloramphenicol used to suppress bacterial growth?

Fungal Identification is rather vague, and no proof of identification presented.

Were the fungi recovered sporulating?

Details difficult to follow with statement such as: (ITS sequencing) (not all isolates); In many instances only till the stage of differentiation between yeasts and moulds.

Results

I did not find the results easy to follow.

Fungi were dominant in both sand and sea water; this is surprising as other studies report yeasts in high numbers from sea water. See the following paper

Communities of culturable yeasts and yeast‑like fungi in oligotrophic hypersaline coastal waters of the Arabian Gulf surrounding Qatar

Rashmi Fotedar  · Mark Chatting · Anna Kolecka · Aisha Zeyara · Amina Al Malki · Ridhima Kaul · Sayed J. Bukhari · Mohammed Abdul Moaiti · Eric J. Febbo · Teun Boekhout · Jack W. Fell They report High yeast counts (>1000 cells/L) were observed in sites 2 (1st sampling) and 13 (4th sampling) which showed a high prevalence of the clinically relevant, and hydrocarbon assimilating species C. tropicalis and C. parapsilos.

Again, sloppy formatting of fungal names: Penicillia (Penicillium?), Fusarium species, the latter word should not be in italics.

Discussion

Poorly formatted and selective in its cover.

Summary

1.The extensive two year study concentrating on exploring the mycobiota in sand and water of the Israeli Mediterranean Sea coast revealed a great diversity of presence of yeast and mold species, primarily in sand and partially also in water. Surely one cannot say there was great diversity in water from the results presented.

4. The data presented in this study may be a basis for regulatory measures to control the level of existence of fungal mycobiota in sand and seawater. As someone with experience of regulator measures, this is fanciful to say the least. You will need much greater details for such an evaluation.

5. Such regulatory measures are recommended for the benefit of public health.

No evidence is presented of cause and effect in this study, so again a rather fanciful statement to make.

Please see more comments in the attachment

Author Response

  1. All references are relevant
  2. Legends to figures were added
  3. The authors believe that the conclusions are supported by the results
  4. We performed english editing
  5. We performed a spell check
please find attached the "Detailed Response"

Reviewer 2 Report

There is an overall benefit to publishing this interesting work, but the text must be revised by a researcher-English native speaker.

Please find in the attch. file with the suggested corrections. 

Author Response

As suggested by the reviewer, we performed english editing.

Reviewer 3 Report

No true fungal pathogens (dermatophytes, Histoplasma capsulatum which is reported in Israel) or others recommended to be reported by Sabino et al. (in addition to those  Cryptococcus neoformans/gattii, Cladiphialophora bantiana) were detected, might be underlined in conclusion.

Line 103: Bruker has 3 recommended protocoles for moulds, direct unsporulated colony, transfer into water from unsporulated colony and colony grown in Sabouraud dextrose broth. It would be interesting to know which one/ones  were used.

Line 244: "Both Aspergillus species may act as allergens inducing respiratory hypersensitivity." Please provide reference.

Line 255 comments on acquired azole resistance of A.fumigatus. This study does not contain resistance data. Previous notes on resistance (C.guilliermondii) is about intrinsic resistance. Please clarify your point ( sand may be a reservoir for resistant isolates?, etc ) or exclude.

Please provide stand-alone figure and table legends as they should be understandable without referring the text.

Please check the spelling and italicisation of microorgnism names in text, figures and tables. Some were marked in yellow.

Author Response

We did a spell check and checked the references

Round 2

Reviewer 2 Report

The article has improved a lot. 

Author Response

The requested change was made. 

The word "phenotypic" was deleted